# New Bisabolane-Type Sesquiterpenoids from *Curcuma longa* and Their Anti-Atherosclerotic Activity

**DOI:** 10.3390/molecules28062704

**Published:** 2023-03-16

**Authors:** Yu-Qin Guo, Guang-Xu Wu, Cheng Peng, Yun-Qiu Fan, Lei Li, Fei Liu, Liang Xiong

**Affiliations:** 1State Key Laboratory of Southwestern Chinese Medicine Resources, Chengdu University of Traditional Chinese Medicine, Chengdu 611137, China; 2School of Pharmacy, Chengdu University of Traditional Chinese Medicine, Chengdu 611137, China; 3Institute of Innovative Medicine Ingredients of Southwest Specialty Medicinal Materials, School of Pharmacy, Chengdu University of Traditional Chinese Medicine, Chengdu 611137, China

**Keywords:** *Curcuma longa* L., bisabolane-type sesquiterpenoids, absolute configuration, anti-inflammatory activity, macrophage, foam cell formation

## Abstract

To explore the sesquiterpenoids in *Curcuma longa* L. and their activity related to anti-atherosclerosis. The chemical compounds of the rhizomes of *C. longa* were separated and purified by multiple chromatography techniques. Their structures were established by a variety of spectroscopic experiments. The absolute configurations were determined by comparing experimental and calculated NMR chemical shifts and electronic circular dichroism (ECD) spectra. Their anti-inflammatory effects and inhibitory activity against macrophage-derived foam cell formation were evaluated by lipopolysaccharide (LPS) and oxidized low-density lipoprotein (ox-LDL)-injured RAW264.7 macrophages, respectively. This study resulted in the isolation of 10 bisabolane-type sesquiterpenoids (**1**–**10**) from *C. longa*, including two pairs of new epimers (curbisabolanones A–D, **1**–**4**). Compound **4** significantly inhibited LPS-induced nitric oxide (NO), interleukin-1β (IL-1β), interleukin-6 (IL-6), tumor necrosis factor-α (TNF-α), and prostaglandin E2 (PGE2) production in RAW264.7 cells. Furthermore, compound **4** showed inhibitory activity against macrophage-derived foam cell formation, which was represented by markedly reducing ox-LDL-induced intracellular lipid accumulation as well as total cholesterol (TC), free cholesterol (FC), and cholesterol ester (CE) contents in RAW264.7 cells. These findings suggest that bisabolane-type sesquiterpenoids, one of the main types of components in *C. longa*, have the potential to alleviate the atherosclerosis process by preventing inflammation and inhibiting macrophage foaming.

## 1. Introduction

Atherosclerosis (AS), a common cardiovascular disease, is an important cause of death and morbidity in the developed world [1,2]. Recently, studies have shown that AS is mainly caused by disorders of lipid metabolism and inflammation [3], so lipid metabolism intervention and inflammation intervention are theoretically the two main means of preventing it. Currently, the modulation of lipid metabolism mainly depends on statins, but these drugs have many adverse effects [4]. Therefore, it is necessary to develop more effective treatment strategies and drugs.

As previously reported, blood-activating herbs have special advantages for the treatment of atherosclerotic cardiovascular diseases and have attracted widespread attention [5]. *Curcuma longa* L., a common medicine for promoting blood circulation and removing blood stasis, has been widely used in traditional Chinese medicine (TCM) for the treatment of hypertension, coronary disease, and peripheral vascular lesions [6,7]. Modern studies have shown that it has anti-inflammatory, vasodilatory, and hypolipidemic activities [8,9], and its main active components are terpenoids and curcumins [10]. However, previous research has mainly focused on curcumin.

In recent years, our group has conducted chemical and pharmacological studies on the terpenoids of *C. longa* and obtained several sesquiterpenoids with novel structures and significant vasodilatory activity [11,12,13], which indicated that the sesquiterpenoids of *C. longa* deserve further study. Thus, the present study continued the phytochemical investigation of *C. longa* and explored the anti-inflammatory activity and inhibitory activity against foam cell formation in RAW264.7 macrophages. Interestingly, 10 bisabolane-type sesquiterpenoids with similar structures were isolated, including two pairs of new epimers (**1**–**4**) and a pair of diastereoisomers (**5** and **6**) (Figure 1).

## 2. Results

### 2.1. Chemical Structure Elucidation

Curbisabolanone A (**1**) was isolated as a colorless oil. Its molecular formula was established as C_15_H_24_O_3_ by an ion peak at *m*/*z* 275.1620 [M + Na]^+^ in the HRESIMS (calcd. for C_15_H_24_O_3_Na, 275.1623). The IR spectrum of compound **1** exhibited characteristic absorption bands for hydroxy (3295 cm^−1^) and carbonyl (1703 and 1665 cm^−1^) functionalities, and its UV absorption maximum at 231 nm showed the possibility of the existence of an, *α*,*β*-unsaturated ketone moiety [14]. In the ^1^H NMR spectrum of compound **1** (Table 1), the typical signals of four methyl groups [*δ*_H_ 0.91 (3H, d, *J* = 6.6 Hz), 1.20 (6H, s), 1.93 (3H, s)], one olefinic proton (*δ*_H_ 5.74), and several aliphatic methylenes and methines between *δ*_H_ 1.74 and 2.64 ppm were observed. The ^13^C NMR (Table 1) and DEPT spectra of compound **1** exhibited 15 carbon resonances for 4 × CH_3_, 4 × CH_2_, 3 × CH (one olefinic), and 4 × C (one olefinic, one oxygenated, and two carbonyl carbons). The above data suggested that compound **1** was a bisabolane-type sesquiterpenoid [15,16]. The HMBC correlations of H_3_-15 with C-2, C-3, and C-4; of H-7 with C-1, C-5, C-6, C-8, and C-9; of H_2_-8 with C-6, C-7, C-9, and C-10; of H_3_-13 with C-10, C-11, and C-12; and of OH-11 with C-10, C-11, C-12, and C-13, combined with the cross peaks of H_2_-4/H_2_-5/H-6/H-7/H_2_-8 and H_3_-14/H-7 in the ^1^H–^1^H COSY spectrum confirmed that the planar structure of **1** was bisabol-2-en-11-ol-1,9-dione (Figure 2).

Curbisabolanone B (**2**) had the same molecular formula as compound **1** based on its HRESIMS. The ^13^C and ^1^H NMR data (Table 1) of compound **2** were very similar to those of compound **1**, which implied that compounds **1** and **2** could be epimers. This deduction was further verified by the HSQC, HMBC, and ^1^H-^1^H COSY data analyses (Figure 2).

Curbisabolanone C (**3**) and Curbisabolanone D (**4**) were assigned the molecular formula of C_15_H_22_O_2_ by HRESIMS. The ^1^H, ^13^C, and DEPT NMR data (Table 2) of compound **3** suggested that it was the 10,11-dehydration product of **1**, which was confirmed by HMBC correlations of H_3_-12 and H_3_-13 with two olefinic carbons (C-10 and C-11) (Figure 2). A comparison of the NMR data of compounds **4** and **3** suggested that they were isomers with the same planar structure. Although this planar structure has been reported in the literature [17], the absolute configurations have not been determined.

Since the relative configurations of compounds **1**−**4** could not be determined by NOESY experiments due to the bond rotation of the side chain, NMR calculations of the chemical shifts and ECD calculations were used to determine their configurations. As for epimers **1** and **2**, NMR calculations at the mPW1PW91/6-311+G(d,p) level were applied to determine their relative configurations based on the PCM model [18]. As shown in Figure 3A,B and Figure 4A,B, the correlation coefficients between the calculated and experimental ^13^C NMR chemical shifts for (6*S*,7*S*)/(6*R*,7*R*) and (6*S*,7*R*)/(6*R*,7*S*) of compounds **1** and **2** were both very close [compound **1**: R^2^ for (6*S*,7*S*)/(6*R*,7*R*): 0.9986, R^2^ for (6*S*,7*R*)/(6*R*,7*S*): 0.9987; compound **2**: R^2^ for (6*S*,7*S*)/(6*R*,7*R*): 0.9986, R^2^ for (6*S*,7*R*)/(6*R*,7*S*): 0.9987]. Thus, the DP4+ probabilities of (6*S*,7*S*)/(6*R*,7*R*) and (6*S*,7*R*)/(6*R*,7*S*) were calculated with the mPW1PW91 function. It can be inferred that (6*S*,7*S*)/(6*R*,7*R*) was the preponderant isomer of compound **1** (Figure 3C) over (6*S*,7*R*)/(6*R*,7*S*), while the relative configuration of compound **2** was the opposite (Figure 4C). Consequently, the absolute configuration of compound **1** could be inferred to be (6*S*,7*S*)/(6*R*,7*R*), and that of compound **2** could be inferred to be (6*S*,7*R*)/(6*R*,7*S*), which were used for ECD calculations to further determine the absolute configurations using the time-dependent density functional theory (TDDFT) method. As shown in Figure 5A,B, the experimental ECD spectra of compounds **1** and **2** matched well with the calculated ECD spectra of (6*S*,7*S*) and (6*R*,7*S*), respectively. Hence, compound **1** was determined to be (+)-(6*S*,7*S*)-bisabol-2-en-11-ol-1,9-dione, while compound **2** was determined to be (–)-(6*R*,7*S*) -bisabol-2-en-11-ol-1,9-dione.

To determine the absolute configurations of compounds **3** and **4**, the same computational methods as those used for compounds **1** and **2** were carried out. Unfortunately, the DP4+ probability did not distinguish compounds **3** and **4**. Therefore, we reviewed the literature to explore methods to solve the absolute configurations of such compounds. From the literature survey, seven pairs of bisabolane-type epimers (Figure 6, **a1**/**a2**–**g1**/**g2**) [16,19,20,21,22,23] structurally similar to compounds **1**−**4** were found, and their optical rotations are listed in Table 3. It was found that the optical rotation is positive when the configuration of C-6 is *S* (6*S*,7*S* or 6*S*,7*R*), while the optical rotation is negative when the configuration of C-6 is *R* (6*R*,7*S* or 6*R*,7*R*). Thus, based on the optical rotations of compounds **3** ([α${]}_{D}^{20}$ +47.6, *c* 0.04, CH_3_OH) and **4** ([α${]}_{D}^{20}$ –46.0, *c* 0.04, CH_3_OH), the absolute configuration of C-6 in compound **3** may be *S* and that in compound **4** may be *R*. To further determine the absolute configuration of C-7 in compounds **3** and **4**, (6*S*,7*S*)-**3**/(6*S*,7*R*)-**3** and (6*R*,7*S*)-**4**/(6*R*,7*R*)-**4** were proposed for ECD calculations. As illustrated in Figure 5C and D, the calculated ECD spectra of (6*S*,7*S*)-**3** and (6*R*,7*S*)-**4** showed good agreement with the experimental curves of compounds **3** and **4**, respectively. Thus, the structure of compound **3** was identified as (+)-(6*S*,7*S*)-bisabol-2,10-dien-1,9-dione, while that of compound **4** was identified as (–)-(6*R*,7*S*)-bisabol-2,10-dien-1,9-dione.

In addition to these new bisabolane-type sesquiterpenoids, six known analogues were isolated from *C. longa* and identified as longpene C [24], longpene D [24], curculonone A [16], 7-dehydroxy-5,9-dioxo-α-bisabolol [25], intermedin B [24], and turmerone Q [26] by comparing their spectroscopic data with those published in the literature.

### 2.2. Physicochemical Properties and Spectroscopic Data of Compounds ***1***–***4***

Curbisabolanone A (**1**): colorless oil; [α${]}_{D}^{20}$ +48.2 (*c* 0.03, CH_3_OH); ECD (MeCN) *λ*_max_ (*Δε*) 191 (+0.54), 231 (0.87), 290 (+0.21), and 344 (+0.10) nm; UV (MeCN) *λ*_max_ (log *ε*) 195 (3.53) and 231 (3.62) nm; IR (KBr) *ν*_max_ 3295, 2959, 2918, 2849, 1703, 1665, 1554, 1441, 1370, 1259, 1207, 1094, 1028, 804, and 695 cm^-1^, (+)-HRESIMS *m*/*z* 275.1620 [M + Na]^+^ (calcd for C_15_H_24_O_3_Na, 275.1623); ^1^H NMR (acetone-*d*_6_, 600 MHz) and ^13^C NMR (acetone-*d*_6_, 150 MHz) data, see Table 1. The original UV, IR, (+)-HR-ESI-MS, ^1^H NMR, ^13^C NMR, DEPT, HSQC, ^1^H-^1^H COSY, and HMBC spectra are shown in Appendix A; see Appendix A.

Curbisabolanone B (**2**): colorless oil; [α${]}_{D}^{20}$ −35.0 (*c* 0.02, CH_3_OH); ECD (MeCN) *λ*_max_ (*Δε*) 194 (−0.69), 228 (−0.84), 296 (+0.07), and 341 (−0.11) nm; UV (MeCN) *λ*_max_ (log *ε*) 190 (3.37) and 231 (3.65) nm; IR (KBr) *ν*_max_ 3295, 2967, 2918, 2871, 1700, 1662, 1565, 1444, 1370, 1262, 1210, 1091, 1033, 868, and 797 cm^−1^, (+)-HRESIMS *m*/*z* 275.1621 [M + Na]^+^ (calcd for C_15_H_24_O_3_Na, 275.1623); ^1^H NMR (acetone-*d*_6_, 600 MHz) and ^13^C NMR (acetone-*d*_6_, 150 MHz) data, see Table 1. The original UV, IR, (+)-HR-ESI-MS, ^1^H NMR, ^13^C NMR, DEPT, HSQC, ^1^H-^1^H COSY, and HMBC spectra are shown in Appendix A; see Appendix A.

Curbisabolanone C (**3**): colorless oil; [α${]}_{D}^{20}$ +47.6 (*c* 0.04, CH_3_OH); ECD (MeCN) *λ*_max_ (*Δε*) 192 (+0.07), 214 (−0.16), and 240 (+0.77) nm; UV (MeCN) *λ*_max_ (log *ε*) 193 (3.08) and 234 (3.40) nm; IR (KBr) *ν*_max_ 2967, 2934, 2876, 1664, 1623, 1444, 1379, 1312, 1207, 1022, and 868 cm^−1^, (+)-HRESIMS *m*/*z* 257.1516 [M + Na]^+^ (calcd for C_15_H_22_O_2_Na, 257.1517); ^1^H NMR (CDCl_3_, 600 MHz) and ^13^C NMR (CDCl_3_, 150 MHz) data, see Table 2. The original UV, IR, (+)-HR-ESI-MS, ^1^H NMR, ^13^C NMR, DEPT, HSQC, ^1^H-^1^H COSY, and HMBC spectra are shown in Appendix A; see Appendix A.

Curbisabolanone D (**4**): colorless oil; [α${]}_{D}^{20}$ −46.0 (*c* 0.04, CH_3_OH); ECD (MeCN) *λ*_max_ (Δ*ε*) 197 (−0.53), 225 (−1.51), and 243 (+0.66) nm; UV (MeCN) *λ*_max_ (log *ε*) 234 (3.73) nm; IR (KBr) *ν*_max_ 2964, 2931, 2873, 1664, 1620, 1444, 1375, 1259, 1210, 1113, 1039, and 865 cm^−1^, (+)-HRESIMS *m*/*z* 257.1516 [M + Na]^+^ (calcd for C_15_H_22_O_2_Na, 257.1517); ^1^H NMR (CDCl_3_, 600 MHz) and ^13^C NMR (CDCl_3_, 150 MHz) data, see Table 2. The original UV, IR, (+)-HR-ESI-MS, ^1^H NMR, ^13^C NMR, DEPT, HSQC, ^1^H-^1^H COSY, and HMBC spectra are shown in Appendix A; see Appendix A.

### 2.3. NMR Data and ECD Calculation

The details of NMR data and ECD calculation of compounds **1**–**4** are shown in Appendix A and Appendix A; see Appendix A.

### 2.4. Cell Viability

After 24 h of treatment with compounds **1**–**8** at concentrations of 100, 50, and 25 μM, RAW264.7 cells did not show a significant reduction in cell viability compared with the serum-free DMEM-treated group. Thus, the anti-inflammatory and anti-macrophage foaming activities of these compounds in RAW264.7 cells were not induced by their cytotoxicity.

### 2.5. Anti-Inflammatory Activity

To evaluate the anti-inflammatory activity of compounds **1**–**8**, a model of LPS-induced inflammation in RAW264.7 macrophages were used. Compounds **2**–**5**, **7**, and **8** inhibited LPS-induced NO production in RAW264.7 cells in a dose-dependent manner compared to the model group (Figure 7A). Compound **4** had the highest inhibition rates at 25, 50, and 100 μM (EC_50_ = 55.40 ± 14.01 μM) (Figure 7B). However, the NO inhibitory effects of all these compounds were weaker than those of the positive control (curcumin, EC_50_ = 12.50 ± 1.30 μM). Additionally, to further elucidate the anti-inflammatory effect of compound **4**, some inflammatory factors were measured. Figure 8 shows that compound **4** at 25, 50, and 100 μM showed a significant reduction in the levels of TNF-α, IL-6, IL-1β, and PGE2 compared with the LPS group.

### 2.6. Anti-Macrophage Foaming Activity

RAW264.7 cells were exposed to 75 μg/mL ox-LDL to induce intracellular lipid accumulation, and they were quantitatively analyzed by Nile Red staining. As shown in Figure 9A, ox-LDL induced a significant increase in the lipid content in RAW264.7 cells compared with the untreated control group (*p* < 0.01), whereas such an abnormal increase could be significantly inhibited by compound **4**. The anti-macrophage foaming activity of compound **4** was also assessed by Oil Red O staining. After treatment with compound **4** at 25, 50, and 100 μM, the degree of cellular foaminess decreased significantly in the ox-LDL-treated RAW264.7 cells (Figure 9B,C). Moreover, compound **4** significantly decreased intracellular TC, FC, and CE levels in RAW264.7 cells at concentrations of 25, 50, and 100 μM compared with the model group (Figure 10).

## 3. Discussion

AS is the leading cause of cardiovascular diseases such as myocardial infarction and coronary artery disease. Growing evidence indicates that AS is an active inflammatory process accompanied by lipid infiltration and the repair of endothelial cell damage. During the lipid infiltration, macrophages phagocytose ox-LDL and turn into foam cells [27]. Foam cells further migrate to the subintima and secrete inflammatory mediators, such as TNF-α, IL-1β, and IL-6, which aggravate local inflammation in the intima and promote the development of AS, eventually leading to the formation and rupture of AS plaques [28,29]. Therefore, inhibiting inflammation, regulating lipid metabolism, and preventing foam cell formation can effectively suppress the development of AS [30].

Bisabolane-type sesquiterpenoids of *C. longa* have been demonstrated to be able to exert anti-inflammatory effects through the NF-κB/MAPK pathway [15,31] and vasorelaxant activities through the PI3K/Akt pathway [11]. However, there is no experimental evidence for the use of such compounds in the prevention and treatment of AS. In this study, 10 structurally similar bisabolane-type sesquiterpenoids were obtained from the ethyl acetate extract of *C. longa*. The biological activities of compounds **1**–**8** were further evaluated in terms of cell viability, cell foaminess, and cell inflammation in RAW264.7 cells. The results showed that compound **4** observably reduced the levels of inflammatory factors (NO, TNF-α, IL-1β, IL-6, and PGE2) and intracellular lipids (TC, FC, and CE) in RAW264.7 cells. Therefore, bisabolane-type sesquiterpenoids of *C. longa* may have anti-AS potential, and the mechanism may be related to anti-inflammatory factors, regulation of lipid metabolism, and reduction of macrophage foaminess. However, due to the limited sample quantities of the isolated natural compounds, more in-depth experimental exploration was not conducted.

Interestingly, the structures of compounds **1**–**8** are very similar, but they differ greatly in terms of anti-inflammatory activity. As depicted in Figure 11, the configuration is an important factor affecting the anti-inflammatory activity. A comparison of compounds **3**/**4** and **5**/**6** indicated that the activity was greatly attenuated when the absolute configuration of C-6 was transferred from *R* to *S*. In addition to configuration, substitution is another important aspect that affects anti-inflammatory activity. Concretely, a comparison of the activities of compounds **4** and **5** indicated that the introduction of an OH group at C-4 resulted in a significant loss of the anti-inflammatory effect. However, by comparing compounds **4** and **8**, it was found that the transfer of the carbonyl group from C-1 to C-4 and the inversion of the C-7 configuration did not affect the intensity of the activity. In addition, the oxidation degree of the side chain may have an impact on such activity. The dehydration of OH-11 in compound **2** to form a double bond [Δ^10(11)^] in compound **4** resulted in a significant increase in the effect. In summary, compound **4** exhibited the most potent anti-inflammatory activity in this study, and the anti-inflammatory activity of the bisabolane-type sesquiterpenoids may be related to the absolute configuration of C-6, the substituent groups, and the degree of oxidation.

## 4. Materials and Methods

### 4.1. General Experimental Procedures

HRESIMS spectra were measured using a Q Exactive instrument (Thermo Scientific™, Waltham, MA, USA). NMR spectra were acquired by a Bruker Avance NEO-600 NMR spectrometer (Bruker Corporation, Billerica, MA, USA) with TMS as an internal standard. The UV and ECD spectra were recorded on an Applied photophysics Chirascan and Chirascan-plus circular dichroism spectrometer (Applied Photophysics Ltd., Leatherhead, UK). IR spectra were recorded using a spectrum one FY-IR spectrometer instrument (Perkin Elmer Inc., Waltham, MA, USA). Optical rotations were measured using an Anton Paar MCP 200 automatic polarimeter (Anton Paar GmbH, Graz, Austria). HPLC separations were carried out using an Agilent 1100 instrument (Agilent Technologies Inc., Santa Clara, CA, USA) equipped with a Welch Ultimate XB-C18 column (10 × 250 mm^2^, 5 μm). Silica gel (200–300 mesh, Yantai Institute of Chemical Technology, Yantai, China) and polyamide (60–90 mesh, Changfeng Chemical Co., Ltd., Jiangsu, China) were used for column chromatography. TLC was performed using silica gel GF_254_ plates (Qingdao Marine Chemical Inc., Qingdao, China). Absorbance was measured with a Molecular Devices SpectraMax iD3 Microplate Reader (Molecular Devices, Sunnyvale, CA, USA). The cells were observed with a Leica DMi1 inverted microscope (Leica Microsystems GmbH, Wetzlar, Germany) for staining. Cell viability was detected by the Cell Counting Kit-8 (CCK-8) (Biosharp Biotechnology, Anhui, China) reagent. Lipopolysaccharide (LPS) and oxidized low-density lipoprotein (ox-LDL) were purchased from SIGMA-ALDRICH and Yiyuan Biotechnology, respectively. Nitric oxide (NO) content was measured using the 2500T NO detection kit (Beyotime Biotechnology, Shanghai, China), while interleukin-6 (IL-6), interleukin-1β (IL-1β), tumor necrosis factor-α (TNF-α), and prostaglandin E2 (PGE2) were measured by the ELISA kit (Elabscience Biotechnology Co., Ltd., Wuhan, China). Intracellular lipid relative contents were determined using Nile Red (Biosharp Biotechnology, Anhui, China) and Oil Red O (Service Biotechnology, Wuhan, China) reagents. The total cholesterol (TC) and free cholesterol (FC) were measured by the ELISA kit (Shanghai Keshun Biotechnology Co., Ltd., Shanghai, China).

### 4.2. Plant Material

The rhizomes of *Curcuma longa* L. (Zingiberaceae) were purchased from Sichuan Neautus Traditional Chinese Medicine Co., Ltd. (Chengdu, China). The sample was identified by Dr. Jihai Gao (Chengdu University of Traditional Chinese Medicine, Chengdu, China). A voucher specimen (No. CL-20160803) was deposited at the Institute of Innovative Medicine Ingredients of Southwest Specialty Medicinal Materials at Chengdu University of Traditional Chinese Medicine.

### 4.3. Extraction and Isolation

The rhizomes of *C*. *longa* (50 kg) were soaked in 95% ethanol overnight and extracted with eight volumes of 95% ethanol under reflux for 3, 2, and 1.5 h, consecutively. Afterward, the EtOH extract was suspended in H_2_O and partitioned successively with petroleum ether and EtOAc to afford a petroleum ether portion of 2 kg and an EtOAc portion of 3 kg. Sequentially, the EtOAc portion underwent silica gel column chromatography and was eluted with petroleum ether and EtOAc (petroleum ether:EtOAc = 1:0, 7:3, and 4:6) and EtOAc and MeOH (EtOAc:MeOH = 1:0, 1:1, and 0:1) to yield six fractions (A–F).

Fraction B was subjected to a silica gel column using a gradient elution with CH_2_Cl_2_ and EtOAc (CH_2_Cl_2_:EtOAc = 1:0–0:1) to yield 16 fractions (B_1_–B_16_). Fraction B_5_ was separated by RP-MPLC and eluted sequentially with a gradient elution system (10–100% MeOH in H_2_O) to obtain 10 fractions (B_5-1_–B_5-10_). B_5-1_ was further separated on a sephadex LH-20 gel column (petroleum ether:CH_2_Cl_2_:CH_3_OH = 5:5:1) to afford B_5-1-1_–B_5-1-5_. B_5-1-3_ was divided into B_5-1-3-1_–B_5-1-3-3_ using the same method as described for B_5-1_. Purification of B_5-1-3-1_ by preparative TLC (petroleum ether:Me_2_CO = 1:2) and reverse-phase semipreparative HPLC (55% MeOH in H_2_O) in turn yielded compound **10** (2.5 mg, *t*_R_ = 62.0 min). Fraction B_7_ was further separated via RP-MPLC with a gradient solvent system (MeOH in H_2_O, 20–100%) to obtain B_7-1_–B_7-16_. Fraction B_7-7_ was separated via silica gel column chromatography with a gradient elution (CH_2_Cl_2_:EtOAc = 50:1–1:1) to obtain four fractions (B_7-7-1_–B_7-7-4_). B_7-7-1_ was further purified by RP semipreparative HPLC (86% MeOH in H_2_O) to yield compound **7** (5.6 mg, *t*_R_ = 28.7 min). Fraction B_9_ was further separated via RP-MPLC with a gradient solvent system (40–100% MeOH in H_2_O) to obtain subfractions B_9-1_–B_9-18_. B_9-7_ was separated into B_9-7-1_–B_9-7-3_ by a sephadex LH-20 gel column (petroleum ether:CH_2_Cl_2_:CH_3_OH = 5:5:1). Subsequently, B_9-7-2_ was purified by preparative TLC (CH_2_Cl_2_:EtOAc = 25:1), followed by reverse-phase semipreparative HPLC (55% MeOH in H_2_O), to yield compound **9** (1.3 mg, *t*_R_ = 72.6 min).

Fraction C was chromatographed on a polyamide column using a gradient solvent system (40–100% EtOH in water) to yield four fractions (C_1_–C_4_). C_1_ was subjected to column chromatography over silica gel eluted with petroleum ether and CH_2_Cl_2_ (petroleum ether:CH_2_Cl_2_ = 200:1–1:1) in a step manner to afford eight fractions (C_1-1_–C_1-8_). Fraction C_1-2_ was further purified by preparative TLC (petroleum ether:Me_2_CO = 3:1), followed by reverse-phase semipreparative HPLC (75% MeOH in H_2_O), to yield compounds **3** (5.4 mg, *t*_R_ = 39.1 min), **4** (2.5 mg, *t*_R_ = 36.5 min), and **8** (1.7 mg, *t*_R_ = 42.3 min). C_1-8_ was divided into four fractions (C_1-8-1_–C_1-8-4_) by RP-MPLC using a gradient solvent system (10–100% MeOH in H_2_O). Fraction C_1-8-3_ was further separated into 10 fractions (C_1-8-3-1_–C_1-8-3-10_) by silica gel column chromatography (petroleum ether:Me_2_CO = 200:1–1:1). Subsequently, fraction C_1-8-3-4_ was chromatographed via preparative TLC developed by petroleum ether and Me_2_CO (petroleum ether:Me_2_CO = 2:1) to obtain C_1-8-3-4-1_–C._1-8-3-4-3_. Further purification of C_1-8-3-4-1_ via RP semipreparative HPLC (65% MeOH in H_2_O) yielded compounds **1** (2.1 mg, *t*_R_ = 33.6 min) and **2** (1.5 mg, *t*_R_ = 30.6 min). Similarly, C_1-8-3-5_ was selected for further purification by preparative TLC (petroleum ether:Me_2_CO = 1.8:1) and reverse-phase semipreparative HPLC (63% MeOH in H_2_O), yielding compounds **5** (1.3 mg, *t*_R_ = 67.0 min) and **6** (0.5 mg, *t*_R_ = 70.8 min).

### 4.4. Cell Lines and Culture

Mouse monocyte-macrophage RAW264.7 was obtained from ATCC, Manassas, VA, USA, and cultured in Dulbecco’s Modified Eagle Medium (DMEM) supplemented with 10% fetal bovine serum at 37 °C, 95% humidity*,* and 5% CO_2_.

### 4.5. Cell Counting Kit-8 Assay

RAW264.7 cells were seeded in 96-well plates at a density of 2 × 10*^5^*/mL and treated with various concentrations of compounds. After incubation for 24 h, 10 μL of CCK-8 solution was added to each well, and the cells were incubated for 1 h. The absorbance was measured at 450 nm using a Molecular Devices SpectraMax iD3 Microplate Reader.

### 4.6. Inflammatory Factor Assay

RAW264.7 cells were plated at a volume of 1 mL per well (2 × 10^5^/mL) into 24-well plates, incubated for 24 h, and then randomly grouped. The serum-free medium group was used as the control group. The model group was treated with 1 μg/mL of lipopolysaccharide (LPS) alone. The administered groups were treated with 1 μg/mL of LPS combined with different concentrations of compounds (25, 50, and 100 μM), and 25 μM curcumin was used as the positive control. The cell supernatants from each group were collected after the cells were incubated for 24 h. NO levels in the cell supernatants were measured by a Griess assay, and the IL-6, IL-1β, TNF-α, and PGE2 levels were measured using ELISA kits.

### 4.7. Inhibition of Macrophage Foaming

#### 4.7.1. Fluorometric Determination of Lipid Accumulation by Nile Red

RAW264.7 cells were plated at 5 × 10^4^ cells/well in 96-well plates and randomly divided into the control, model (75 μg/mL ox-LDL), positive control (25 μM simvastatin), and tested compound groups (25, 50, and 100 μM of compound **4**). In each group, the cells were incubated for 24 h and fixed for 20 min by 4% paraformaldehyde successively. After the plates were washed with phosphate buffered saline (PBS), the cells were incubated with a 1 μg/mL Nile Red solution for 30 min. The fluorescence was measured at an excitation wavelength of 530 nm and an emission wavelength of 590 nm.

#### 4.7.2. Oil Red O Staining

RAW264.7 cells were plated at 5 × 10^5^ cells/well in 24-well plates and randomly divided into groups as described above. After a 24 h treatment, the Oil Red O staining was performed according to the instructions. Then, the Oil Red O was dissolved with isopropanol, and the absorbance was measured at 550 nm.

#### 4.7.3. Intracellular Lipid Content Assay

RAW264.7 cells were seeded into 6-well plates at a density of 1 × 10^6^/mL and grouped as described above. After a 24 h treatment, the contents of total cholesterol (TC) and free cholesterol (FC) in each group of cells were measured based on the kit instructions. The cholesterol ester (CE) content was calculated as follows: CE = TC − FC.

### 4.8. Statistical Analysis

Statistical analyses were performed with *SPSS 25.0*. Data are presented as the mean ± standard deviation (SD). The EC_50_ value was analyzed from the cumulative concentration-effect curves by non-linear regression analysis. A one-way ANOVA was used for comparisons between groups. *p* < 0.05 was considered to be statistically significant.

## 5. Conclusions

In summary, 10 bisabolane-type sesquiterpenoids, including two new pairs of epimers (**1**–**4**), were obtained from *C. longa* in this study. Compounds **2**, **3**, **4**, **5**, **7**, and **8** exhibited anti-inflammatory effects, with compound **4** being the most active. Further studies on compound **4** found that it significantly reduced the levels of NO, IL-1β, IL-6, TNF-α, and PGE2 in RAW264.7 cells. In addition, compound **4** showed an anti-macrophage foaming effect and significantly reduced the intracellular TC, FC, and CE levels in RAW264.7 cells. These results suggest that bisabolane-type sesquiterpenoids, one of the main types of components of *C. longa*, have the potential to alleviate the AS process by preventing inflammation and inhibiting macrophage foaming.

## Figures and Tables

**Figure 1 molecules-28-02704-f001:**
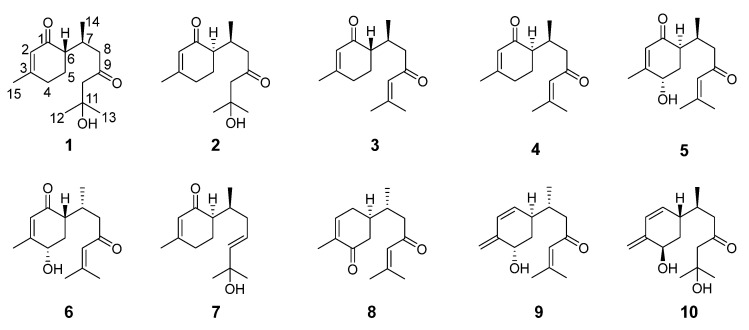
Chemical structures of compounds **1**–**10.**

**Figure 2 molecules-28-02704-f002:**
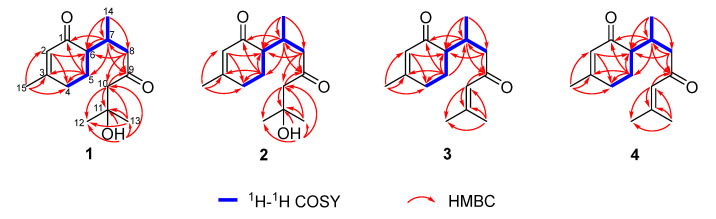
Key ^1^H–^1^H COSY and HMBC correlations of compounds **1**–**4**.

**Figure 3 molecules-28-02704-f003:**
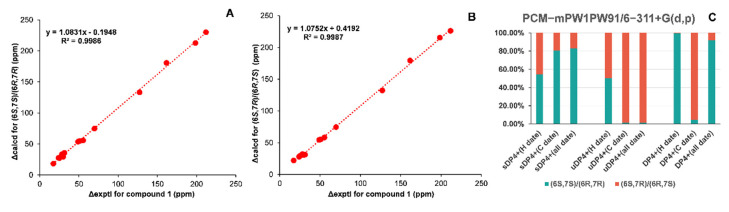
Comparison of the experimental and calculated NMR chemical shifts of compound **1**. (**A**) Regression analysis of experimental ^13^C NMR chemical shifts of compound **1** vs. calculated ^13^C NMR chemical shifts of (6*S*,7*S*)/(6*R*,7*R*)-**1**. (**B**) Regression analysis of experimental ^13^C NMR chemical shifts of compound **1** vs. calculated ^13^C NMR chemical shifts of (6*S*,7*R*)/(6*R*,7*S*)-**1**. (**C**) DP4+ probability for (6*S*,7*S*)/(6*R*,7*R*)-**1** and (6*S*,7*R*)/(6*R*,7*S*)-**1** at the PCM-mPW1PW91/6-311+G(d,p) level.

**Figure 4 molecules-28-02704-f004:**
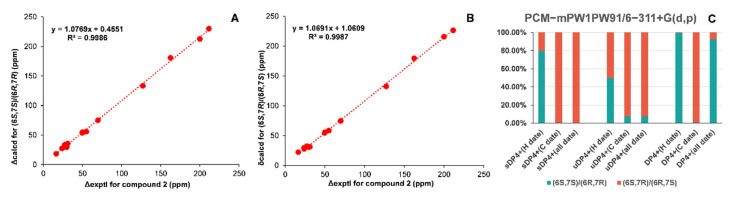
Comparison of the experimental and calculated NMR chemical shifts of compound **2**. (**A**) Regression analysis of experimental ^13^C NMR chemical shifts of compound **2** vs. calculated ^13^C NMR chemical shifts of (6*S*,7*S*)/(6*R*,7*R*)-**2**. (**B**) Regression analysis of experimental ^13^C NMR chemical shifts of compound **2** vs. calculated ^13^C NMR chemical shifts of (6*S*,7*R*)/(6*R*,7*S*)-**2**. (**C**) DP4+ probability for (6*S*,7*S*)/(6*R*,7*R*)-**2** and (6*S*,7*R*)/(6*R*,7*S*)-**2** at the PCM-mPW1PW91/6-311+G(d,p) level.

**Figure 5 molecules-28-02704-f005:**
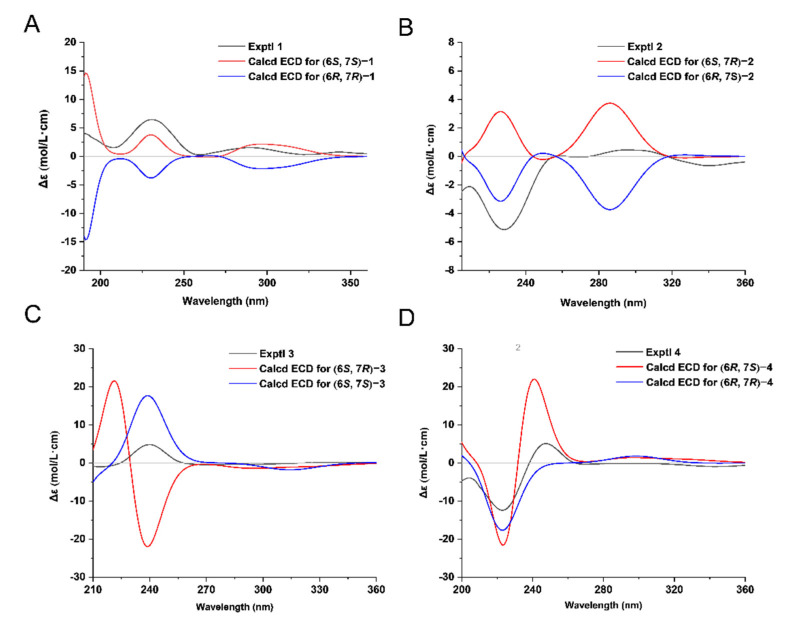
Experimental and calculated ECD spectra of (**A**) compound **1**, (**B**) compound **2**, (**C**) compound **3**, and (**D**) compound **4**.

**Figure 6 molecules-28-02704-f006:**
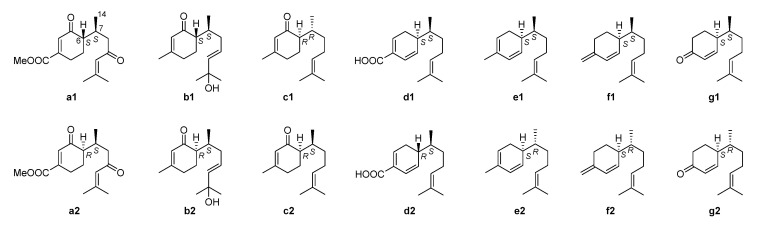
Chemical structures of seven pairs of bisabolane epimers.

**Figure 7 molecules-28-02704-f007:**
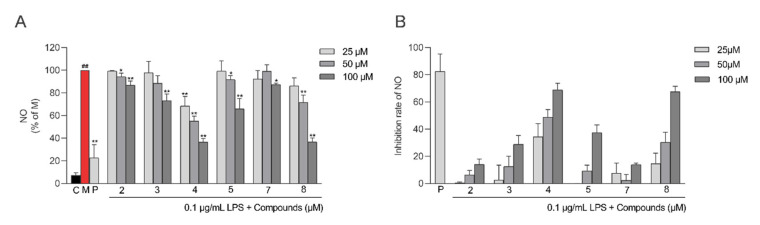
Inhibitory effects of compounds **2**–**5, 7**, and **8** on lipopolysaccharide (LPS)-induced NO production in RAW264.7 cells. (**A**) Relative NO content in RAW264.7 cells of each group. (**B**) The inhibition rates of LPS-induced NO production by the compounds (mean ± SD, *n* = 3). Statistical analysis: **^##^** *p* < 0.01 vs. the control group; * *p* < 0.05; and ** *p* < 0.01 vs. the LPS-treated model group. C: control group, M: model group, and P: positive group.

**Figure 8 molecules-28-02704-f008:**
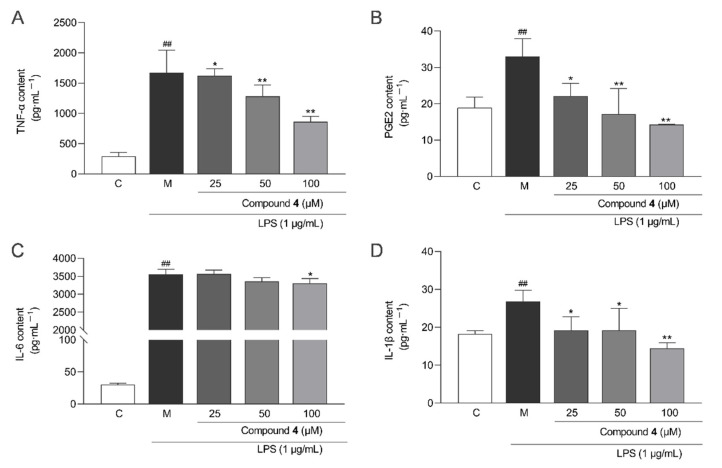
Inhibitory effect of compound **4** on the contents of (**A**) tumor necrosis factor-α (TNF-α), (**B**) prostaglandin E2 (PGE2), (**C**) interleukin-6 (IL-6), and (**D**) interleukin-1β (IL-1β) in LPS-treated RAW264.7 cells (mean ± SD, *n* = 3). Statistical analysis: **^##^** *p* < 0.01 vs. the control group; * *p* < 0.05; and ** *p* < 0.01 vs. the LPS-treated model group. C: control group; M: model group.

**Figure 9 molecules-28-02704-f009:**
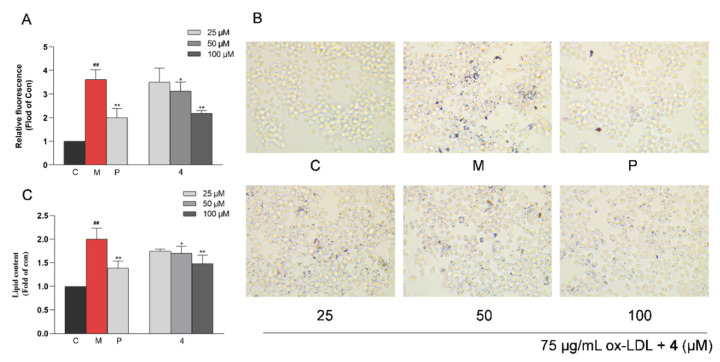
Inhibitory effect of compound **4** on intracellular lipid accumulation in RAW264.7 cells. (**A**) Intracellular lipid contents were measured by the fluorescence intensity at an excitation/emission wavelength of 530/560 nm. (**B**) Oil Red O staining images of RAW264.7 cells (magnification 400×). (**C**) Quantitative Oil Red O content at 550 nm. Data are expressed as the mean ± SD (*n* = 3). Statistical analysis: **^##^** *p* < 0.01 vs. the control group; * *p* < 0.05; and ** *p* < 0.01 vs. the ox-LDL-treated model group. C: control group, M: model group, and P: positive group.

**Figure 10 molecules-28-02704-f010:**
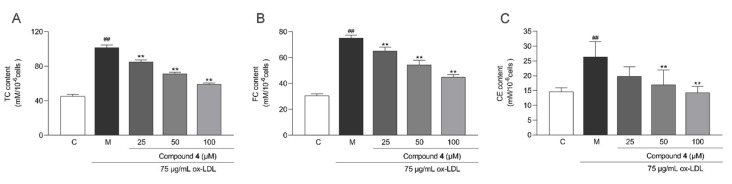
Inhibitory effect of compound **4** on the intracellular contents of (**A**) total cholesterol (TC), (**B**) free cholesterol (FC), and (**C**) cholesterol ester (CE) in RAW264.7 cells (mean ± SD, *n* = 3). Statistical analysis: **^##^** *p* < 0.01 vs. the control group; and ** *p* < 0.01 vs. the ox-LDL-treated model group. C: control group; M: model group.

**Figure 11 molecules-28-02704-f011:**
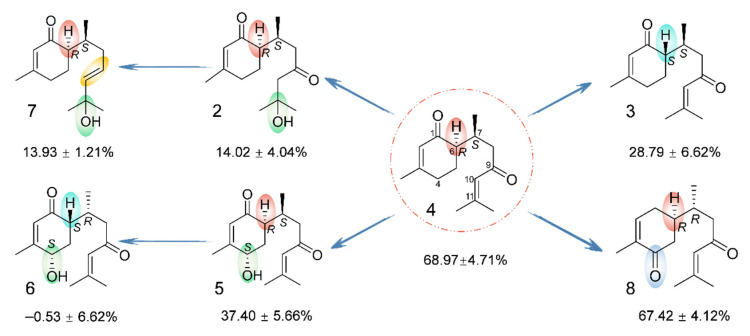
Summary of structure–activity relationships of bisabolane-type sesquiterpenoids. The numbers below the compounds indicate the maximum inhibition rates of the compounds against LPS-induced NO production.

**Table 1 molecules-28-02704-t001:** ^1^H (600 MHz) and ^13^C NMR (150 MHz) data for compounds **1** and **2** in acetone-*d*_6_ (*δ* in ppm, *J* in Hz).

No.	1		2	
	*δ* _H_	*δ* _C_	*δ* _H_	*δ* _C_
1		198.6		199.9
2	5.74 s	127.3	5.73 brs	127.2
3		161.7		162.4
4	2.37 m	31.5	2.35 m	31.1
5	(a) 2.00 m	25.2	(a) 1.96 m	23.8
	(b) 1.74 m		(b) 1.74 m	
6	2.17 dt (12.6, 4.2)	51.0	2.15 dt (12.6, 4.2)	50.2
7	2.64 m	28.3	2.79 m	27.3
8	(a) 2.51 dd (16.2, 3.6)	49.0	(a) 2.58 dd (17.4, 6.0)	49.8
	(b) 2.37 m		(b) 2.46 dd (17.4, 7.8)	
9		211.9		211.5
10	2.60 s	55.5	(a) 2.60 d (15.0)	55.0
			(b) 2.56 d (15.0)	
11		69.9		69.9
12	1.20 s	29.8	1.19 s	29.8
13	1.20 s	29.8	1.20 s	29.8
14	0.91 d (6.6)	17.4	0.79 d (7.2)	16.5
15	1.93 s	24.0	1.92 brs	24.0
OH-11	3.84 s		3.87 s	

**Table 2 molecules-28-02704-t002:** ^1^H (600 MHz) and ^13^C NMR (150 MHz) data for compounds **3** and **4** in CDCl_3_ (*δ* in ppm, *J* in Hz).

No.	3		4	
	*δ* _H_	*δ* _C_	*δ* _H_	*δ* _C_
1		200.8		200.7
2	5.83 s	127.2	5.82 s	126.9
3		161.6		161.6
4	(a) 2.32 dd (7.8, 4.2)	31.1	(a) 2.30 m	30.5
	(b) 2.32 dd (7.8, 4.2)		(b) 2.30 m	
5	(a) 2.00 m	24.7	(a) 1.97 m	23.2
	(b) 1.77 m		(b) 1.81 m	
6	2.19 m	50.5	2.12 m	49.6
7	2.65 m	28.8	2.79 m	27.9
8	(a) 2.55 m	48.6	(a) 2.41 m	49.3
	(b) 2.19 dd (15.0, 4.2)		(b) 2.35 m	
9		201.2		200.8
10	6.16 s	124.1	6.10 brs	123.9
11		155.3		155.6
12	1.88 s	27.8	1.87 brs	27.9
13	2.14 s	20.9	2.12 brs	20.9
14	0.93 d (6.6)	17.5	0.87 d (6.6)	16.8
15	1.93 s	24.2	1.92 s	24.3

**Table 3 molecules-28-02704-t003:** Optical rotations of compounds **a1**/**a2**−**g1**/**g2**.

Compounds	Absolute Configuration	Optical Rotations
**a1** [19]	(6*S*,7*S*)	[α${]}_{D}^{19}$: +72.7 (*c* 0.33, EtOH)
**a2** [19]	(6*R*,7*S*)	[α${]}_{D}^{17}$: −34.3 (*c* 0.33, EtOH)
**b1** [16]	(6*S*,7*S*)	[α${]}_{D}^{25}$: +58.7 (*c* 0.11, CHCl_3_)
**b2** [16]	(6*R*,7*S*)	[α${]}_{D}^{25}$: −38.4 (*c* 0.13, CHCl_3_)
**c1** [20]	(6*R*,7*R*)	[α${]}_{D}^{20}$: −20.0 (*c* 1.20, CHCl_3_)
**c2** [20]	(6*R*,7*S*)	[α${]}_{D}^{20}$: −40 (*c* 1.1, CHCl_3_)
**d1** [21]	(6*S*,7*S*)	[α${]}_{D}^{25}$: +88 (*c* 0.1, CH_2_Cl_2_)
**d2** [21]	(6*R*,7*S*)	[α${]}_{D}^{25}$: −50(*c* 0.1, CH_2_Cl_2_)
**e1** [22]	(6*S*,7*S*)	[α${]}_{D}^{25}$: +175.4 (*c* 0.41, CHCl_3_)
**e2** [22]	(6*S*,7*R*)	[α${]}_{D}^{25}$: +128.6 (*c* 0.98, CHCl_3_)
**f1** [23]	(6*S*,7*S*)	[α${]}_{D}^{22}$: +5.95 (*c* 1.17, CHCl_3_)
**f2** [23]	(6*S*,7*R*)	[α${]}_{D}^{22}$: +39.58 (*c* 0.43, CHCl_3_)
**g1** [22]	(6*S*,7*S*)	[α${]}_{D}^{25}$: +80.6 (*c* 1.78, CH_2_Cl_2_)
**g2** [22]	(6*S*,7*R*)	[α${]}_{D}^{25}$: +60.2 (*c* 0.41, CH_2_Cl_2_)

## Data Availability

The data presented in this study are available in the Appendix A or can be provided by the authors.

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
