# Peer review of "New Bisabolane-Type Sesquiterpenoids from Curcuma longa and Their Anti-Atherosclerotic Activity"

_molecules, 2023, doi:10.3390/molecules28062704_

Round 1

Reviewer 1 Report

In the article "New bisabolane-type sesquiterpenoids from Curcuma longa and their anti-atherosclerotic activity" the authors Guo et al. describe a series of new compounds isolated from Curcuma longa. The authors first justify their line of research, the isolated compounds, structure elucidation and biological activity tests. The methods used are adequate and described  comprehensively. The conclusions are clear and well justified, and nicely show how that line of research could be followed up. 

Over the past years it has become more common to provide the raw/processed data via public repositires, instead of printouts of spectra and other information. I would suggest that the authors consider using this as a better way to provide the supplemental material. 

Author Response

A word version of the Supplemental Material has been reuploaded. In this Supplemental Material, all NMR spectra can be viewed directly by a Mestrenova software.

Reviewer 2 Report

The article shows total coherence in all its parts. The effort that the authors made to try to identify the configuration of compounds 3 and 4 is commendable. Additionally, the finding of new compounds makes the study of interest to readers of the journal and to researchers seeking alternative treatments to synthetic drugs. 

I believe that there was considerable and rigorous work behind this text that leaves no doubt of the results which are important and should be extended not only in vitro but also scaled up to in vivo experiments in mice.

Author Response

Comment 1

I believe that there was considerable and rigorous work behind this text that leaves no doubt of the results which are important and should be extended not only in vitro but also scaled up to in vivo experiments in mice.

Response

Thank you very much for your good suggestion. It is really true as you suggested that further in vivo mice studies could provide a more comprehensive and accurate interpretation of the pharmacological effects of the compounds. Unfortunately, we are unable to further investigate the effects using the in vivo mice models due to the limited sample quantities of the isolated natural products. In the future we will make more efforts to get more quantities to do the animal experiments.

Reviewer 3 Report

The researchers studied the composition of the rhizomes of Curcuma longa, resulting in the isolation of 10 bisabolane-type sesquiterpenoids (110), including two pairs of new epimers (14). The structures of new compounds were elucidated through the analysis of NMR spectral data and extensive comparison with the literature data in combination of NMR chemical shifts and ECD calculations. In the bioassay experiments, compound 4 exhibited significant anti-inflammatory and anti-macrophage foaming effects. These results suggest that bisabolane-type sesquiterpenoids of C. longa could be developed as drug leas for the treatment of atherosclerotic cardiovascular diseases. Based on these intriguing findings, this manuscript will gain extensive interests from readers globally.

I will recommend it to be accepted after minor revision.

1. Please give the trial names for the new compounds 14.

2. The first paragraph of the subsection ‘2.1. Chemical Structure Elucidation’ should be deleted.

3. The styles of some citations in the main text such as on P3L9 and the first paragraph on P7 are required to be revised.

4. It was interesting to find that the configurations of C-6 in the molecules 1 and 2 were different, but the coupling constants of H-6 were the same. Could you explain it?

5. Please add tR values for the compounds purified by HPLC on P13.

Author Response

Comment 1

Please give the trial names for the new compounds 14.

Response

Thank you for your suggestion. We have added the trial names of compounds 14 in the revised manuscript.

Comment 2

The first paragraph of the subsection ‘2.1. Chemical Structure Elucidation’ should be deleted.

Response

We have deleted this redundant paragraph.

Comment 3

The styles of some citations in the main text such as on P3L9 and the first paragraph on P7 are required to be revised.

Response

Thank you for your comment. We have corrected the citation formatting in the revised manuscript.

Comment 4

It was interesting to find that the configurations of C-6 in the molecules 1 and 2 were different, but the coupling constants of H-6 were the same. Could you explain it?

Response

Thank you very much for your professional comment. Interestingly, the coupling constants of H-6 in (6S,7S)-1 and (6R,7S)-2 are the same (dt, J = 12.6, 4.2 Hz). Analysis of the dominant conformations of (6S,7S)-1 and (6R,7S)-2 (Figure R1) found that the same coupling constants were related to different adjacent protons. Specifically, in compound 1, H-6 is in an axial orientation, and H-5a and H-5b are equatorial-oriented and axial-oriented, respectively, in the six-membered ring. Thus, the coupling constants of H-6 with H2-5 are J5a,6 = Jeq,ax = 4.2 Hz and J5b,6 = Jax,ax = 12.6 Hz. The coupling constants of H-6 with H-7 is small (J6,7 = 4.2 Hz) due to a dihedral angle of about 60 degree.

In compound 2, H-6 is in an equatorial orientation, and H-5a and H-5b are equatorial-oriented and axial-oriented, respectively, in the six-membered ring. Thus, the coupling constants of H-6 with H2-5 are J5a,6 = Jeq,eq = 4.2 Hz and J5b,6 = Jax,eq = 4.2 Hz. The coupling constants of H-6 with H-7 is large (J6,7 = 12.6 Hz) due to a dihedral angle of about 180 degree. 

Comment 5

Please add tR values for the compounds purified by HPLC on P13.

Response

Thank you for your suggestion. We have added the tR values of the isolated compounds in the revised manuscript.
